# The Dietary Management of Patients with Irritable Bowel Syndrome: A Narrative Review of the Existing and Emerging Evidence

**DOI:** 10.3390/nu11092162

**Published:** 2019-09-09

**Authors:** Joost Algera, Esther Colomier, Magnus Simrén

**Affiliations:** 1Department of Internal Medicine and Clinical Nutrition, Sahlgrenska Academy, University of Gothenburg, 413 45 Gothenburg, Sweden; 2Department of Gastroenterology and Hepatology, Amsterdam UMC (location VUmc), VU University Amsterdam, 1081 HV Amsterdam, The Netherlands; 3Department of Translational Research for Gastrointestinal Disorders, University Hospital KU Leuven, University of Leuven, 3000 Leuven, Belgium

**Keywords:** irritable bowel syndrome, dietary management, low FODMAP diet, gluten-free diet, dietary fiber, lactose-free diet, exclusion diets

## Abstract

Even though irritable bowel syndrome (IBS) has been known for more than 150 years, it still remains one of the research challenges of the 21st century. According to the current diagnostic Rome IV criteria, IBS is characterized by abdominal pain associated with defecation and/or a change in bowel habit, in the absence of detectable organic causes. Symptoms interfere with the daily life of patients, reduce health-related quality of life and lower the work productivity. Despite the high prevalence of approximately 10%, its pathophysiology is only partly understood and seems multifactorial. However, many patients report symptoms to be meal-related and certain ingested foods may generate an exaggerated gastrointestinal response. Patients tend to avoid and even exclude certain food products to relieve their symptoms, which could affect nutritional quality. We performed a narrative paper review of the existing and emerging evidence regarding dietary management of IBS patients, with the aim to enhance our understanding of how to move towards an individualized dietary approach for IBS patients in the near future.

## 1. Introduction

With a prevalence of approximately 10% worldwide, irritable bowel syndrome (IBS) is one of the most common gastrointestinal (GI) disorders [1,2]. Patients with IBS do not have readily identifiable underlying structural abnormalities, but the diagnosis is made based on the current diagnostic standard, the Rome IV criteria, as: Recurrent abdominal pain, on average, at least 1 day/week, associated with 2 or more of the following criteria: related to defecation; associated with a change in frequency of stool; and associated with a change in form (appearance) of stool. The criteria should be fulfilled for the last 3 months with a symptom onset at least 6 months before the diagnosis [3]. Based on the dominant stool form or consistency, IBS is also subtyped into IBS with constipation (IBS-C), IBS with diarrhea (IBS-D), mixed IBS (IBS-M), and unspecified IBS (IBS-U) [4,5]. The syndrome is one of the leading causes for consultations in gastroenterology outpatient clinics, as well as in primary care, and the most common reason for referral to gastroenterology clinics. Symptoms interfere with the daily life of many patients, reduce health-related quality of life and lower the work productivity [5].

IBS is currently described as a disorder of disturbed gut–brain interactions, with a heterogeneous and incompletely understood pathophysiology. Altered gut-brain interactions, visceral hypersensitivity [6], psychosocial distress, and gastrointestinal motor disturbances are considered to be of importance for IBS or at least subsets of patients [5]. Moreover, over the past years, the number of factors that contribute to the pathophysiology has expanded. Intestinal immune activation, increased intestinal permeability [7], an altered microbiome [8], and food hypersensitivity [9] are examples of other factors that can contribute to symptoms in subsets of IBS patients (Figure 1). The heterogeneity of IBS, even within individual subtypes, makes it difficult to design a treatment algorithm to fit all patients. The lack of a thorough understanding of pathophysiological mechanisms has hampered the development of effective treatments, but also has led to a symptom-directed treatment approach, rather than primarily focusing on the underlying pathophysiology [10,11,12].

The majority of patients with IBS report that intake of food leads to generation of symptoms or worsening of symptoms [13,14]. Moreover, it has also been demonstrated that ingestion of a meal can provoke symptoms in patients with IBS [9,15]. Hence, there has been an increase in the interest in dietary management of symptoms in IBS, especially during the last decade. A survey among 1562 gastroenterologists in the United States stated that over half of the physicians recommended diets to 75% of their IBS patients [16]. Although the relative importance of different components of meals for symptom generation in IBS is still unclear, different diets are increasingly implemented as a valid and effective treatment option in the clinical setting [17]. However, the influence of nutrient triggers on the generation of symptoms in IBS already appeared in the literature in the 1980s [18,19,20]. In this narrative review we aim to discuss how dietary approaches used for the management of IBS patients have evolved over time and led to the dietary guidelines used in clinical practice today. Moreover, the importance of a more personalized dietary treatment approach for IBS patients is now frequently advocated, which is likely to be more common in the near future.

## 2. Methodology

We conducted a PubMed search with the following key words to identify previously written reviews on the subject and relevant studies performed in humans: ‘irritable bowel syndrome AND diet’, ‘irritable bowel syndrome AND elimination diet’, ‘irritable bowel syndrome AND dietary fiber’, ‘irritable bowel syndrome AND NICE guidelines’, ‘irritable bowel syndrome AND FODMAP’, ‘irritable bowel syndrome AND lactose’, ‘irritable bowel syndrome AND fructose AND fructan’, ‘irritable bowel syndrome AND low-carbohydrates’, and ‘irritable bowel syndrome AND gluten’. Afterwards searches were limited to articles in the English language. Additionally, we performed a hand search with articles about irritable bowel syndrome and diets that were written before the year 1990. All searches were completed by June 2019.

## 3. History of Dietary Management of IBS: The Bran Era and Exclusion Diets

In 1977, the first randomized controlled trial (RCT) involving a dietary intervention in IBS patients was published, where addition of wheat fiber in IBS was assessed [21]. The low-fiber diets consumed in the Western countries were held responsible for the high prevalence of IBS. In the literature, the late seventies and early eighties are referred to as the ‘bran era’ [20]. Bran is the hard outer layer of grains, which contains high dietary fiber, essential fatty acids, and significant quantities of protein, vitamins, and dietary minerals. An increase in fiber content by the addition of wheat bran to food has been proposed to ameliorate symptom generation in IBS [22]. Ingestion of bran accelerates slow transit and delays rapid intestinal transit, suggesting that it should lead to a more regular bowel habit. Not only IBS symptoms, but symptoms originating from many emerging diseases of the Western world at the time such as atherosclerosis, obesity, colon cancer, diverticular disease, diabetes mellitus, and gall stones were also considered to be related to a fiber deficiency, and it was proposed that increasing intake of bran would be beneficial. The importance of the placebo effect was not underestimated and in some cases bran was described as a less toxic placebo than many drugs. Undoubtedly, it helped certain IBS patients, and in particular IBS patients with constipation. However, after the initial optimism, it was reported that a large proportion of patients with IBS actually reported an exacerbation of symptoms when treated with bran or a high-fiber diet, and an excessive consumption of bran was even thought to create patients with IBS [23,24]. This controversy and the limited evidence for a direct beneficial effect of high-fiber diets caused researchers to believe that the beneficial effects of these diets probably resulted from the displacement of nutrients such as fat and a reduction in energy intake. However, the bran era created awareness of the importance of nutrition and cleared the way for other dietary treatment options in IBS [20].

During the same time period, there was also an interest in the relevance of specific food intolerances in patients with IBS. Jones et al. first investigated foods that provoked symptoms in IBS patients, and found that wheat, corn, dairy products, coffee, tea, and citrus fruits were found to be of relevance for food intolerance in IBS patients [19]. All patients found to be intolerant to wheat had a jejunal biopsy, and in all subjects the biopsy was histologically normal. After ingestion of the triggering factors, patients demonstrated a significant increase of rectal prostaglandin E2, and in a subset of the cases rectal PGE2 correlated with wet faecal weight and the symptoms could be provoked by the food items during double-blind tests. This led researchers to conclude that food intolerances may be a major pathogenetic factor in the IBS, which in turn led to the development of exclusion diets as a possible treatment option. However, the detection of specific food intolerances is cumbersome, requiring strict exclusion diets, followed by double-blind placebo-controlled challenges with the respective food items. During the following years some research groups were able to confirm the relevance of food intolerance in IBS [25], whereas several other groups failed to reach the same conclusion [19,26,27]. In the studies, patients with predominant constipation consistently failed to respond to an exclusion diet, which then led to the hypothesis that the subgroup characterized by diarrhea might be most likely to respond to dietary manipulations.

In the early 1990s, guidelines providing dietary information for physicians treating IBS patients appeared (Figure 2) [28,29]. The primary advice was to include a prospective dietary history with detailed information about symptoms. Based on a food diary, a detailed dietary assessment (at least 7 days) should include the quality and quantity of foods consumed. The patient should record all symptoms that occur after meals or that are potentially linked to meal ingestion. Additionally, a description of the frequency and consistency of bowel movements should be included. Instead of eliminating every possible nutrient trigger, the physician could then use this information and may detect a relationship between symptoms and ingestion certain foods or a combination of foods. However, this approach has not been formally tested in prospective clinical trials, but is based on clinical experience.

## 4. IgG Elimination Diet

As an alternative to the strict exclusion diets previously discussed, an elimination diet based on the presence of IgG antibodies to foods has been proposed. In the first intervention study using this approach, the patients were assessed for presence of immunoglobulin (Ig) G antibodies towards food items. Thereafter they were randomized to receive a diet that eliminated food products to which they had elevated levels of IgG antibodies, or a sham diet that eliminated the same number of foods but not those to which they had antibodies. After 3 months a 10% greater reduction in the IBS symptom severity score was seen in the patients that followed the individualized elimination diet compared to patients allocated to the sham diet. Furthermore, when participants reintroduced eliminated foods, symptoms worsened in the patients with the individualized elimination diet based on presence of IgG antibodies [30]. Subsequently, there have been uncontrolled studies confirming the positive effects of an individualized IgG elimination diet and detected improvements in IBS symptoms and quality of life in general [31,32]. Even though these results were promising, the last studies on this topic were published in 2006, and the use of this approach in clinical settings today is limited. Furthermore, there are some concern about the replicability of the results and also regarding influence of differences in food consumption patterns. Further studies are needed to assess the relevance of food IgG antibodies in symptom generation in IBS patients.

## 5. Dietary Fiber

As previously discussed, the bran era resulted in conflicting results, indicating that dietary fibers could both improve and worsen symptoms of IBS patients. Dietary fibers are non-digestible carbohydrates (e.g., cellulose, resistant starch and glucans), which form key structural materials of cereals, fruits, vegetables and legumes. Fibers can be divided into two main groups, based on water solubility: soluble and insoluble fibers. In the GI tract, soluble fibers form a gel that interacts with gut bacteria and can shorten GI transit. The bacteria produce active metabolites such as short chain fatty acids, which interact with inflammatory pathways. Thus, short chain fatty acids link the gut microbiome with the metabolic profile of the host. In contrary, insoluble fibers barely change in the GI tract. Besides water solubility, several other chemical and physical characteristics of fiber foods influence the gut physiology, such as fermentability, viscosity and bulking/binding capacity. Previous studies have demonstrated that high intake of insoluble fibers increases water content and fecal bulking, resulting in accelerated GI transit time. This is a likely explanation of the benefit of high dietary fiber intake in IBS patients with predominant constipation (IBS-C). However, the largest problem with fiber intake is the formation of gas, which may lead to bloating, abdominal distension and flatulence, but this problem seems to be less prominent with soluble than with insoluble fibers [33,34,35,36].

Moreover, the systematic review and meta-analysis of Moayyedi et al. studied 14 RCTs and found a significant benefit of dietary fibers in global symptom improvement in all IBS patients. The benefit was only seen in studies that used soluble fibers as intervention, and no beneficial effect was seen for insoluble fibers. Furthermore, they found no safety issues or harmful effects of dietary fibers, and no significant heterogeneity between results of the studies was found. However, there were some limiting factors of the studies e.g., small samples, variations in duration of the therapy and no clear IBS diagnosis since the Rome criteria was only used in 2 trials. Nevertheless, they concluded that soluble fibers could be recommended to IBS patients [37].

The recommendation of solely soluble fibers has its limitations, since the distinction of fibers in terms of insoluble and soluble has been proposed to be partly outdated. Both soluble and insoluble fibers frequently co-exist in intact cell-walls of plants, and the physiological responses of the gut can differ independent of solubility [38]. De Vries et al. compared 3 different fiber groups (fruits, cereals and vegetables), independent of solubility, and their effect on fecal weight and GI transit time. Fermentability was included in the assessment of the fiber groups. Less fermentable fibers contributed most to the total fecal weight, especially cereals (oats, rice bran, whole-grain pasta and whole-grain bread) and vegetables. All fiber groups reduced transit time in individuals with a gut transit time of more than 48 h. The fiber groups did not influence transit time in individuals with gut transit times of less than 48 h, indicating that fibers can normalize delayed transit, but not further accelerate normal transit times [39].

Taken together, dietary fibers seem to be beneficial in patients with IBS, especially in patients with IBS-C. Dietary fibers are also safe and inexpensive, since they are widely available in different foods. Therefore, IBS-C patients should be encouraged to take a wide variety of fiber foods, which are less fermentable, usually found in different cereals, and if gas-related symptoms is a problem, intake of soluble rather than insoluble fibers seems beneficial.

## 6. The NICE Guidelines

Recommendation for the initial approach of IBS in primary care has been presented in the National Institute for Health and Care Excellence (NICE) guidelines from the United Kingdom. Figure 3 shows a summary of the proposed clinical work-up of patients with IBS, including dietary advice, derived from the NICE guidelines of IBS [40].

These guidelines recommend primary care physicians to provide IBS patients with information that explains the importance of self-management. This includes encouragement to improve general lifestyle factors such as healthy eating habits, increase physical activity, and to follow simple dietary advice, such as regularly eating meals, thoroughly chewing meals, avoiding missing meals and sufficiently drinking fluids. Restricting recommendations include limiting certain dietary factors, such as caffeine, alcohol, spicy foods and fatty food, which IBS patients frequently report to worsen GI symptoms [14]. These dietary factors influence the GI tract via different mechanisms, and potentially also in different anatomical regions. Caffeine can cause GI symptoms not only due to increased gastric acid secretion, but also via enhanced colonic and rectosigmoid motor activity [41]. Chronic alcohol consumption influences absorption and disturbs GI motility and intestinal permeability, and in vivo and in vitro studies found decrease of immunoreactive neurons in the jejunum of individuals with chronic alcohol consumption [42]. Spicy foods can cause burning sensations and abdominal pain. The bioactive substance in hot peppers is capsaicin which accelerates GI transit via specific receptors and can cause visceral hypersensitivity, which can induce GI symptoms [43]. Fatty foods affect small bowel motility, stimulate the gastrocolonic reflex, and exaggerate visceral hypersensitivity in patients with IBS, which also can lead to worsening of symptoms in IBS [44].

Another recommendation included in the initial dietary approach to patients with IBS is to limit the intake of resistant starch, which reaches the colon undigested, and this can be found in processed foods. Moreover, restricting gas-producing food items such as onions, cabbage, beans, carbonated beverages and artificial sweeteners is often advocated. Intake of fibers should be reviewed early, since the amount and type of dietary fibers may be adjusted, depending on the symptom profile (see above). If these initial strategies to optimize the diet fails to improve the symptoms, primary care physicians can consider referring refractory IBS patients to secondary care, for more advanced dietary management [40].

Recently, there has been an increased scientific and clinical interest in more advanced dietary management strategies in IBS. Since IBS patients relate their symptoms to specific food groups, including foods containing incompletely absorbed carbohydrates, wheat and dairy products [13], recent studies have investigated diets reducing or excluding the intake of these food groups with promising, but partly contradictory results. These will be reviewed in the following sections.

## 7. FODMAPs

A diet specifically developed for the management of IBS is the low Fermentable Oligosaccharides, Disaccharides, Monosaccharides, And Polyols (FODMAP) diet. When the first line dietary management strategy in the previously mentioned NICE guidelines do not lead to an adequate symptom control, a low FODMAP diet is often proposed. FODMAP is a collective term for short-chain carbohydrates that are incompletely absorbed in the small intestine, and includes oligosaccharides including fructans/fructo-oligosaccharides, and galacto-oligosaccharides, lactose, fructose in excess of glucose, and polyols such as sorbitol and mannitol [45]. These carbohydrates then enter the colon where they are fermented, causing production of gas in the lower GI tract [46]. In addition, FODMAPs are osmotically active, leading to an increased water content in the intestinal lumen [47]. Together, these two mechanisms can result in luminal distention, which could lead to symptoms in susceptible individuals, including abdominal pain, diarrhea, flatulence and bloating (Figure 4) [48]. However, FODMAP ingestion does not normally cause GI symptoms in healthy adults. Conversely, the underlying abnormalities in gut physiology and in particular the presence of visceral hypersensitivity might explain why patients with IBS report symptoms after intake of FODMAPs [46,47,49]. At present, these FODMAPs have been identified as some of the most important dietary triggers in IBS patients.

Controlled studies have established the efficacy of a low FODMAP diet that eliminates the intake of foods containing FODMAPs (see below). However, most studies have only addressed the short-term effect (of up to 4 weeks) of this diet. The diet remains complex, can lead to low calorie intake, and requires individualized explanation and follow-up by an experienced dietician [45,50,51]. Moreover, current recommendation is to initially restrict the diet for a short period and then to gradually reintroduce food items rich in FODMAPs in order to identify individual food items/FODMAPs that should be restricted long-term. Hence, the long-term recommendation is to go for a restriction of FODMAPs rather than total elimination, but so far few studies have addressed how efficacious this is [52]. Therefore, future studies addressing the effects of low FODMAP diets, should focus more on the reintroduction phase and assess the long-term effects of the diet.

One of the most influential studies in this area is a randomized controlled clinical cross-over trial that found lower overall symptom scores in patients receiving the low FODMAP diet vs. a typical Australian diet [50]. However, a recent single-blinded RCT indicated that a diet low in FODMAPs reduces symptoms of IBS as well as the traditional dietary advice, i.e., the NICE guidelines mentioned above [51]. Moreover, there is also another recent study comparing a low FODMAP diet with a modified NICE diet in IBS patients with no differences in the primary endpoint between the groups (overall improvement), but with some of the secondary endpoints (specific IBS symptoms) demonstrating larger improvement in the low FODMAP group [53,54]. Table 1 gives an overview of clinical trials assessing the effect of a low FODMAP intervention in IBS patients, but with different study designs and control groups, which needs to be taken into account when assessing the true clinical efficacy of a low FODMAP diet in IBS. Moreover, a recent meta-analysis highlights that there currently is very low-quality evidence that a low FODMAP diet is effective in reducing symptoms in IBS patients. However, among the available dietary intervention studies, the low FODMAP diet has still the greatest evidence for efficacy in IBS [55]. Hence, more data regarding dietary interventions is still needed.

Even though dietary interventions in general are considered low risk, a short-term intervention of a low FODMAP diet has been found to change the colonic microbiome and reduce the concentration of beneficial gut bacteria [56]. Whether this is the case for long-term use of less restrictive FODMAP diets rather than a more strict elimination diet is still unclear. One potential approach that can limit this potentially non-beneficial effect of a low FODMAP diet could be to use a probiotic product together with the low FODMAP diet. This strategy was recently tested and showed to prevent the reduction of beneficial *Bifidobacterium* spp. in the gut [57].

## 8. Lactose

Another dietary factor that many IBS patients avoid is lactose. Lactose is a disaccharide composed of glucose and galactose, and is a FODMAP when it is not digested in the small intestine. It is the main carbohydrate source in mammalian milk. The digestion of lactose takes place in the small intestine and this is genetically regulated. All humans are able to digest lactose in the neonatal phase, but only 25 to 33% keep the ability to digest lactose in adulthood and have the genetic trait of lactase persistence. The rest of the population in the world is not able to digest lactose and have the genetical trait of lactase non-persistence i.e., lactase deficiency. Lactase is an enzyme, localized on the upper surfaces of enterocytes in the small intestine, which hydrolyzes lactose into glucose and galactose. After rapid absorption, glucose is used for energy, and galactose is used as a part of glycoproteins (Figure 5). The prevalence of lactase non-persistence ranges widely across the world, and is dependent on ethnic background, e.g., in Asian countries up to 90% have lactase non-persistence, in African countries 65 to 75%, in Mediterranean countries 40%, and in Central and Northern-European countries 2 to 20% [64].

Individuals are considered to be lactose intolerant when the ingestion of lactose results in symptoms, such as flatulence, bloating, cramps and diarrhea. Individuals with lactase non-persistence and infrequent consumption of dairy may develop lactose intolerance. The main mechanism behind the symptoms is the result of undigested lactose entering the small and large intestine, where it acts as a FODMAP (Figure 4). Recently published reviews provide a comprehensive overview of lactose maldigestion, malabsorption and intolerance, including the pathophysiology, its relation to other disorders and options for treatment [64,65].

Lactose intolerance can sometimes be mistaken for cow’s milk protein allergy, but these disorders have different pathophysiology. The proteins whey and casein are responsible for causing allergic reactions in individuals with cow’s milk protein allergy. The allergic reactions can be both Immunoglobulin E and non-Immunoglobulin E mediated. Systemic responses can occur such as skin lesions, respiratory distress, GI symptoms and anaphylaxis. Compared to lactose intolerance, the GI symptoms of cow’s milk protein allergy are more severe, including rectal blood loss and severe diarrhea [65].

Symptoms of IBS and lactose intolerance overlap and due to this, some researchers (in the second part of the twentieth century) believed that lactose intolerance was the cause of IBS [65], but this theory was later rejected, and it became clear that IBS and lactose intolerance have partly different pathophysiology. Moreover, patients with IBS that are lactase persistent, are able to normally digest lactose. However, since IBS and lactose intolerance both are common conditions, co-occurrence of these disorders are also common, and it is likely that patients with IBS are more sensitive to ingestion of dairy products if they have lactose maldigestion, relative to non-IBS subjects with lactose maldigestion. It is important to consider lactose intolerance before the diagnosis of IBS is made. The golden standard for diagnosing lactose intolerance is to perform a hydrogen breath test after oral intake of lactose (25 to 50 g) [64,65,66].

Several studies, all non-RCTs, have investigated the relation of lactose-free (or dairy-free) diet and IBS. Four studies investigated IBS patients with positive lactose hydrogen breath tests, and evaluated GI symptom scores at baseline and after a period of low lactose or lactose-free diet. The dietary periods ranged from 3 weeks to 5 years [67,68,69,70]. Only one study, with a small number of subjects (*n* = 16) found a significant difference in GI symptom improvement when patients were on a low lactose diet, but did not specify the amount of ingested lactose [68]. Another study found a higher incidence of lactose malabsorption in IBS patients, compared to healthy controls with Northern-European background. However, patients with IBS did not respond better to a low lactose diet (<9 g/day), and no association was found between IBS-type symptoms and lactose intolerance [67].

The studies that have investigated the use of low lactose or lactose-free diet in patients with IBS were recently summarized by the British Dietetic Association. They provided guidelines, which emphasize that it is important to consider lactose intolerance in the diagnostic work-up of patients with suspected IBS, especially in individuals of ethnic backgrounds with higher incidence of lactase non-persistence. There is not sufficient evidence to recommend a low lactose or lactose-free diet to all patients with IBS [71]. The intake of lactose is restricted in individuals who are following a low FODMAP diet, but to focus solely on avoiding lactose may only give minor benefits in most patients with IBS.

## 9. Low-Fructose/Fructan Diet

Over the past three decades, the annual consumption of fructose has dramatically increased [72]. In 2006, Shepherd et al. evaluated the effects of reducing fructose and fructans in IBS patients [58]. Results showed that patients who were adherent to a low fructose diet experienced better symptom response compared to patients that were non-adherent, with greater improvement in GI symptoms, such as abdominal pain, bloating, gas, nausea, diarrhea, and constipation [58]. Similarly, symptom improvements were detected in IBS patients with a known fructose malabsorption when a fructose-restricted diet was maintained [73]. Patients reported improvement in abdominal pain, bloating, belching, and diarrhea compared to baseline. The concept of reducing fructose and fructans has thereafter expanded to also include other poorly carbohydrates in the FODMAP concept, described above. Today, widespread recommendations involving a fructose and fructan-restricted diet outside the FODMAP diet are limited, and just like a low FODMAP diet these dietary approaches have challenges with adherence and long-term maintenance [73].

## 10. Low-Carbohydrate Diet

Recently, there has been a lot of focus on low-carbohydrate diets for weight loss, control of metabolic disease such as diabetes, and exercise performance. Here a moderately lower amount of carbohydrates (between 40 and 50% of the daily energy intake) were compatible with a healthy state and may represent a satisfactory and scientifically-based choice for people with metabolic disorders [74,75,76,77]. Apart from a study published by Austin et al., very little is known about the impact of a low-carbohydrate diet in IBS patients [78]. According to this single study, which only included IBS-D patients, participants had an adequate relief of IBS symptoms for at least 2 weeks during the study. For a period of 4 weeks, they all consumed less than 20 g of carbohydrates per day. Before the 4-week period, participants consumed a diet where approximately 55% of the daily energy intake were from carbohydrates, 30% from fat, and 15% from protein. During the intervention period, approximately 51% of the calories came from fat, 45% from protein, and only 4% from carbohydrates. Abdominal pain, stool consistency/frequency, and quality of life all improved [78]. Additional controlled studies are needed before a low-carbohydrate diet to control symptoms in IBS patients can be recommended.

## 11. Gluten

Wheat grains are built of several components, including different proteins. Gluten is the storage protein of wheat and are made of glutenin and gliadin. Similar proteins are found in barley, rye and oats; hordein, secalin and avenins, respectively. All these proteins are referred to as gluten. Other components of wheat are albumins such as amylase-trypsin inhibitors (ATI’s), and starch, which contains fructans [79]. Fructans are oligosaccharides that are included in the FODMAP concept [80]. Hence, individuals on a wheat-containing diet have intake of gluten proteins, ATI’s and FODMAPs, which are all components that may have a relation with GI symptoms in IBS patients. This makes it difficult to assess the component(s) that are responsible for symptom improvement when a patient excludes wheat from the diet. Figure 6 shows an overview of the components of wheat and their possible relation to GI symptoms in IBS.

Wheat-containing foods are currently responsible for up to 50% of energy intake in all humans [81]. However, the world population has increasingly avoided gluten (wheat) over the last decade. It is estimated that up to 20% of the population of Western countries follows a strict gluten-free diet (GFD) [82]. Causality of symptom generation by gluten is seen in celiac disease. A multidisciplinary task force in Oslo defined celiac disease as ‘a chronic small intestinal immune-mediated enteropathy precipitated by exposure to dietary gluten in genetically predisposed individuals’ [83]. Other gluten-related disorders are wheat allergy, which is Immunoglobulin-E mediated, and non-celiac gluten (or wheat) sensitivity [84]. In contrast to celiac disease, non-celiac gluten sensitivity is a relatively new disease entity with no clear pathophysiological mechanism known so far. An international group of experts have conducted standardized criteria for the diagnosis, since there are no known biomarkers to confirm non-celiac gluten (or wheat) sensitivity. The Salerno Experts’ criteria include a randomized, double-blind, placebo-controlled, gluten challenge where patients are challenged for 1 week with gluten (at least 8 g) and another week with placebo with a washout period separating the interventions. The diagnosis is confirmed when the patient shows an improvement of at least 30% in three main symptoms during the placebo period compared to the gluten period [85]. The patients with a diagnosis of non-celiac gluten (or wheat) sensitivity contribute various symptoms to the intake of gluten (or wheat) including extraintestinal symptoms and GI symptoms (e.g., abdominal pain, bloating and altered bowel habits). From this description, it follows that there is a symptomatic overlap between IBS and non-celiac gluten (or wheat) sensitivity.

Several studies have investigated the efficacy of a gluten-free diet as a treatment for IBS, but with somewhat conflicting and heterogeneous results. In the studies there are similarities, but also methodological differences, which makes them difficult to compare. Table 2 displays studies assessing the relevance of gluten for symptoms in patients with IBS and/or non-celiac gluten (or wheat) sensitivity. Some studies included run-in periods from 2 to 6 weeks where patients already were on a gluten-free diet and/or low FODMAP diet. During the period where the effect of gluten on symptoms was assessed, the daily amount of gluten used in the studies ranged from 4.4 to 52 g, which can be compared with the normal intake of gluten in western countries, which is estimated to be between 5 and 20 g [79]. The majority of studies did not control for dietary confounders, with few exceptions [86]. However, as described previously, excluding wheat-containing foods not only lowers the intake of gluten, but also lowers intake of ATIs and fructans, and all these components may have association with GI symptoms. A possible explanation for the heterogeneous results in the existing studies is that only a subgroup of patients with IBS and/or non-celiac gluten sensitivity clinically respond with reduced symptoms when on a gluten-free diet. A recent meta-analysis concluded that only 16% of non-celiac gluten (or wheat) sensitivity patients have gluten-related symptoms, whereas 40% of these patients have similar symptoms to placebo [87]. Furthermore, a recent study suggested that decrease of dietary fructans rather than gluten are responsible for the symptomatic improvement in patients with non-celiac gluten (or wheat) sensitivity when they are on a gluten-free diet [88].

Taken together, several components of wheat could be responsible for GI symptoms in patients with IBS and non-celiac gluten (or wheat) sensitivity. The pathophysiological mechanism responsible for this is not evident. However, it seems plausible that a subgroup of these patients could benefit from a gluten-free or at least a wheat-free diet. Further studies are needed to understand the role of the specific components of gluten (or wheat) in symptom generation in patients with IBS and non-celiac gluten (or wheat) sensitivity, including search for clinical or biochemical markers that can be used to select patients for this dietary management strategy.

## 12. Future Directions and Recommendations

The relation between food and symptoms in IBS is complex, and over the years many diets have been studied. Figure 7 displays the timeline of the dietary studies that we reviewed in the previous sections, and the change in diagnostic criteria for IBS over time.

Dietary trials are complicated and have their weaknesses and strengths. In general, dietary studies have a small number of subjects, especially compared to pharmacological trials. Additionally, methodological issues, such as inadequate blinding of subjects and dieticians are common. Information about dietary management in IBS is widely accessible, which could make blinding of patients excessively difficult. The awareness of possible effective treatments could also induce a significant placebo response. Moreover, placebo responses are very prevalent in trials with IBS patients in general. Food challenges could also induce physiological responses in individuals without an organic cause, due to the expectation distress (i.e., nocebo effect) [99]. The placebo and nocebo effect in IBS patients point out the importance of the gut-brain axis which acts bidirectionally. To reduce these effects, and to provide high quality evidence, comparative dietary trials should ideally be randomized, double-blind and placebo-controlled. The discovery of FODMAPs shed light on the importance of specific dietary triggers in GI disorders and could have opened the door to dietary management beyond the FODMAP concept. When future trials investigate provoking food components and potential therapeutic dietary interventions, they should provide information on the intake of the food components prior to the intervention and evaluate the level of restriction required [99,100,101].

Another important area of future research is also the potential to predict responses to dietary management. A recent study found promising results, indicating that responsiveness to a low FODMAP diet may be predicted by the bacterial profile in IBS patients before starting the treatment [102]. Even though further studies are needed to evaluate the clinical usefulness of this approach, this study highlights that predictors of response and biomarkers may lead to an optimized individualized dietary management of patients with IBS.

We reviewed the studies that have investigated dietary management in IBS patients during the last five decades, but clinical experience proves that providing dietary recommendations for IBS patients remains difficult for physicians. Not only because of the complexity, necessity of a dietician, heterogeneity and conflicting results of the dietary trials, but also because of unknown long-term effects. At the moment, the scientific interests are focused on the low FODMAP diet, which has shown to be effective in several studies, and the gluten-free diet, which might be effective in a subgroup of IBS patients. Future trials, assessing long-term effects, are needed regarding safety issues. Furthermore, a more obvious multidisciplinary approach in future research, may also further contribute to the understanding of the mechanisms causing symptoms in IBS patients, and lead to development of better and more individualized treatment options in IBS.

## Figures and Tables

**Figure 1 nutrients-11-02162-f001:**
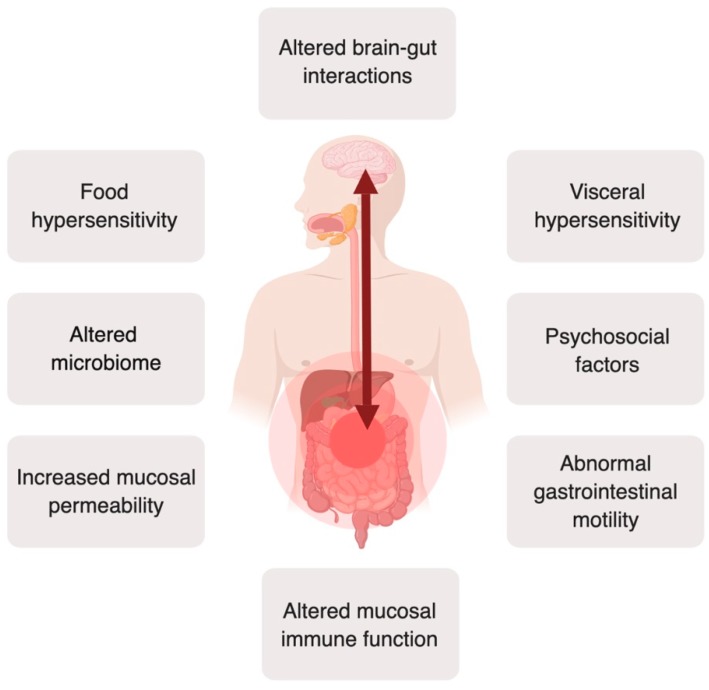
An overview of important pathophysiological factors in IBS. The current view of the heterogeneous IBS pathophysiology is that it is caused by altered brain-gut interactions, with various alterations and abnormalities along the brain-gut axis in subsets of IBS patients. Created with BioRender.

**Figure 2 nutrients-11-02162-f002:**
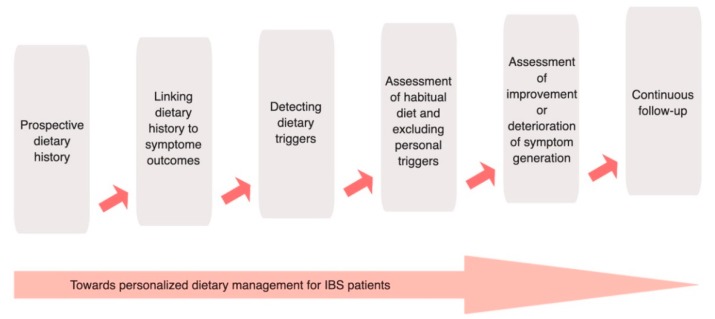
Advise on dietary habits, lifestyle, and how to avoid symptom triggers. Guidelines regarding dietary habits, lifestyle, and avoidance of symptom triggers were already described in early 1990s and were based on clinical experience. This approach can still be used to work towards a personalized dietary management of IBS patients. Abbreviation: IBS: irritable bowel syndrome. Created with BioRender.

**Figure 3 nutrients-11-02162-f003:**
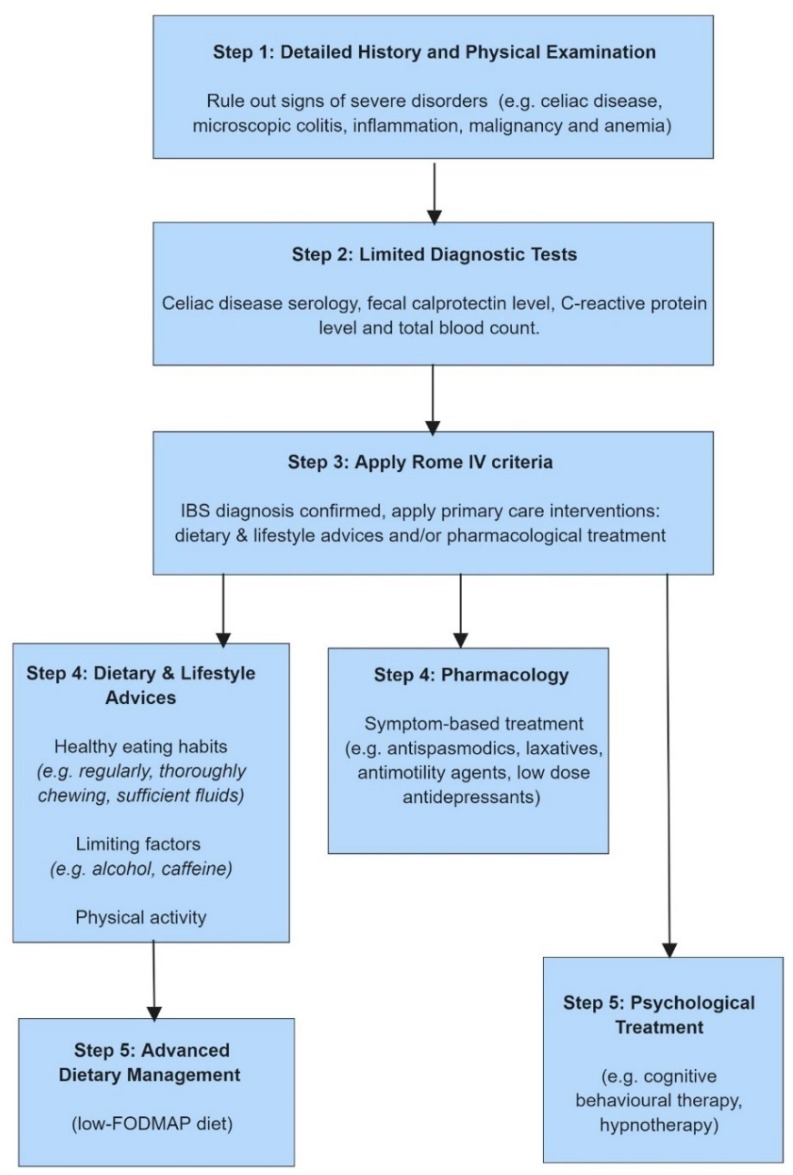
Diagnostic work-up and interventions in patients with IBS, derived from the NICE guidelines. Primary care physicians should apply step 1–4, and can refer patients to secondary care (step 5) if they develop refractory IBS. Abbreviations: FODMAP: fermentable oligo-, di-, monosaccharides and polyols; IBS: irritable bowel syndrome; NICE: National Institute for Health and Care Excellence. Created with BioRender.

**Figure 4 nutrients-11-02162-f004:**
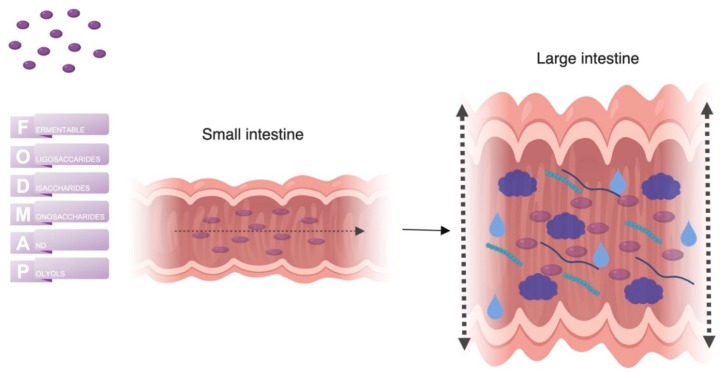
Mechanism of action of FODMAPs. When FODMAPs pass the small intestine where they are incompletely absorbed and can pass into the colon. In the colon the osmotically active short-chain carbohydrates increase the luminal water content. Fermentation of FODMAPs by colonic bacteria causes the production of gas. Increased luminal water content and gas production result in a distention of the large intestine, which in turn could generate GI symptoms. Abbreviations: FODMAPs: Fermentable oligosaccharides, disaccharides, monosaccharides and polyols; GI: gastrointestinal. Created with BioRender.

**Figure 5 nutrients-11-02162-f005:**
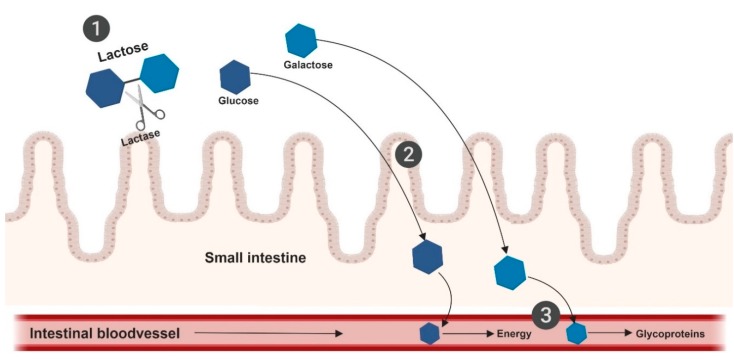
The digestion of lactose in the small intestine. (1) Hydrolyzation of lactose by lactase, located in upper layer of enterocytes. (2) Rapid absorption of monosaccharides, glucose and galactose, which is maximal in the proximal jejunum. (3) Glucose will be used for energy, galactose as a part of glycoproteins. Created with BioRender.

**Figure 6 nutrients-11-02162-f006:**
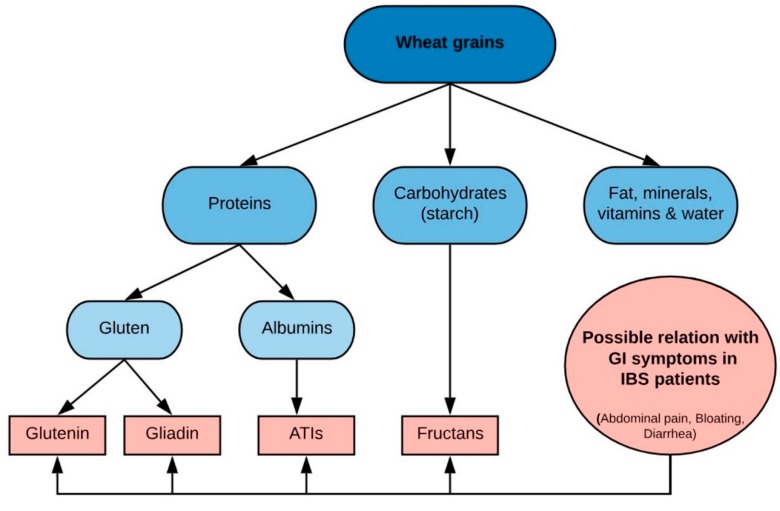
Schematic overview of wheat grain components and relation to gastrointestinal symptoms in IBS. Abbreviations: ATIs: amylase-trypsin inhibitors; GI: gastrointestinal; IBS: irritable bowel syndrome. Created with BioRender.

**Figure 7 nutrients-11-02162-f007:**
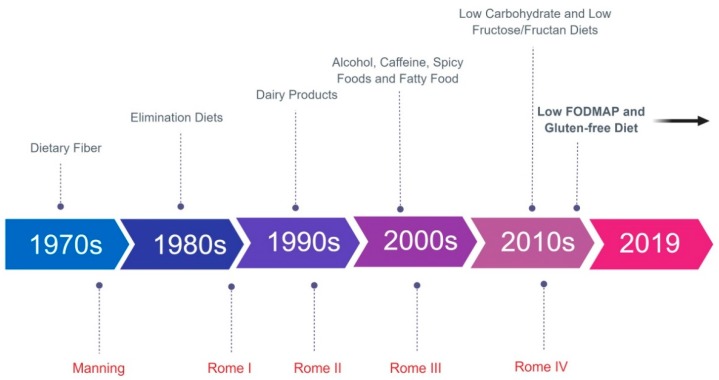
Timeline of dietary trials and diagnostic criteria for irritable bowel syndrome. Abbreviation: FODMAP: fermentable, oligo-, di-, monosaccharides and polyols. Created with BioRender.

**Table 1 nutrients-11-02162-t001:** Clinical trials evaluating the effect of a low FODMAP in patients with IBS.

Study (Year) Country	Design, Population (*n*)	Interventions	Main Findings
Shepherd et al. (2006) Australia [58]	**Single-center** study, IBS patients with fructose malabsorption **(*n* = 62)**	Diet **avoiding free fructose and short-chain fructans**, limitation of the total dietary fructose load, encouragement of foods with balanced amount of fructose/glucose, 40 months	Seventy-four percent responded positively regarding overall abdominal symptoms. This positive response was better in the adherent group compared to the non-adherent group.
Ong et al. (2010) Australia [46]	**Single-center RCT**, IBS patients and HV **(*n* = 30)**	**Low FODMAP** diet vs. **high FODMAP** diet for 2 days with 7-day wash-out period	Higher levels of breath hydrogen were found in HV and IBS patients on a high FODMAP diet. Patient following the high FODMAP diet had more GI symptoms and lethargy. HV receiving the high FODMAP diet only reported more flatulence.
Staudacher et al. (2011) UK [59]	**Single-center Clinical Observational study**, IBS patients **(*n* = 82)**	**Low FODMAP** vs. **standard** dietary advice for IBS patients (based on the NICE guidelines)	Seventy-six percent of the patients on a low-FODMAP diet were satisfied with their symptom response compared to 54% receiving the standard dietary advice. Eighty-two percent reported improvement in bloating with low-FODMAP vs 49% following the NICE guidelines. For 85% and 87% of patients following the low-FODMAP diet abdominal pain and flatulence improved respectively compared to 61% and 50% on the standard diet.
Staudacher et al. (2012) Australia [60]	**Single-center RCT**, IBS patients **(*n* = 41)**	**Low FODMAP** diet vs. **habitual** diet for 4 weeks	Lower intake of fermentable carbohydrates, and lower proportions/concentrations of bifidobacteria was noted in the intervention group compared to the group following their habitual diet. Sixty-eight percent of the patients in the intervention group reported adequate symptom control compared to 23% of the patients with habitual food intake.
de Roest et al. (2013) New Zealand [61]	**Single-center** study, IBS patients **(*n* = 90)**	**Low FODMAP diet**, mean of 15.7 months follow-up	At follow-up, patients reported improvement in abdominal pain, bloating, flatulence and diarrhea. Patients with fructose intolerance experienced an even greater improvement.
Halmos et al. (2014) Australia [50]	**Single-center RCT**, cross-over, IBS patients **(*n* = 30) and healthy controls (*n* = 8)**	**Low FODMAP** vs. **typical Australian** diet for 21 days with a washout period of at least 21 days	Patients on the low FODMAP diet reported improvement of their global IBS symptoms. Abdominal pain, bloating, and passing flatus were significantly better in the low FODMAP group. In most patients, the greatest improvement in symptoms occurred during the first week. Symptoms were minimal and unaltered by either diet among controls.
Böhn et al. (2015) Sweden [51]	**Multicenter RCT**, IBS patients **(*n* = 75)**	**Low FODMAP** diet vs. a **traditional IBS** diet (based on the NICE guidelines) for 4 weeks	During the intervention, the severity of IBS symptoms was reduced in both groups. At the end of the intervention, 50% of the patients on a low-FODMAP diet had a reduction in IBS severity scores (≥50) compared with baseline vs 46% of the patients following the traditional IBS diet.
Eswaran et al. (2016) US [53,54]	**Single-center RCT**, IBS-D patients **(*n* = 92)**	**Low FODMAP** diet vs. a **modified diet based on the NICE guidelines (mNICE)** for 4 weeks	Fifty-two percent of the low FODMAP vs. 41% of the mNICE group reported adequate relief of their IBS-D symptoms, which was not significant. The low FODMAP diet led to significantly greater improvement in individual IBS symptoms, particularly pain and bloating, and quality of life compared with the mNICE diet.
Hustoft et al. (2017) Norway [62]	**Single-center RCT**, IBS-D and IBS-M patients **(*n* = 20)**	**Low FODMAP** diet for 3 weeks & afterwards randomization to a **FODMAP supplement** or **maltodextrin** (placebo) for 10 days with a wash-out period of 3 weeks	Patients receiving the placebo compared to the FODMAP supplement reported a significant relief of symptoms, 80% compared to 30% respectively. After following the low FODMAP diet, alterations in inflammatory cytokines, microbiota profile and SCFAs were detected.
Staudacher et al. (2017) UK [57]	**Two-center RCT**, IBS patients **(*n* = 104)**	**Low FODMAP** diet vs. **sham diet** (restriction of similar amount of foods, but maintaining the FODMAP content in the diet) with randomization to a multi-strain **probiotics** vs. **placebo** for 4 weeks	The low FODMAP diet was associated with an adequate relief of symptoms and a significant reduction of symptom scores compared to placebo, 57% compared to 38% respectively. Co-administration of the probiotic increased the number of Bifidobacterium species compared to placebo.
McIntosh et al. (2017) Canada [63]	**Single-center RCT**, IBS patients **(*n* = 37)**	**Low FODMAP** diet vs. **high FODMAP** diet for 3 weeks	Patients with a low FODMAP intake had a significant improvement in symptom scores and had changes in their metabolome compared to patient following the high FODMAP diet. FODMAPs modulated the microbiota and histamine levels in a subset of patients.

Abbreviations: IBS: irritable bowel syndrome; RCT: randomized controlled trial; HV: healthy volunteers; FODMAP: fermentable oligosaccharides, monosaccharides, and polyols; NICE: National Institute for Health and Care Excellence; IBS-QOL: irritable bowel syndrome-quality of life; SCFAs: short-chain fatty acids.

**Table 2 nutrients-11-02162-t002:** Studies assessing the role of gluten in patients with IBS and/or non-celiac gluten (or wheat) sensitivity.

Study (Year) Country	Design, Population (*n*)	Interventions	Main Findings
Dale et al. (2018) Norway [89]	RDBPC, **cross-over trial** NCGS patients on a GFD **(*n* = 20)**	**GFD vs. GCD**, 4 challenges (2 gluten, 2 placebo) 4 days per intervention, 3 days washout. (muffins with gluten 11 g/day vs. gluten-free muffins)	No significant differences in symptom severity between gluten and placebo challenges. High symptom scores during all challenges.
Skodje et al. (2018) Norway [88]	RDBPC, **cross-over trial** self-reported NCGS patients on GFD >6 months **(*n* = 59)**	**GFD** (placebo) vs. **GCD** (5.7 g/day) **vs. Fructans** (2.1 g/day), 1 week per intervention, 1 week washout. (concealed muesli bars).	Significant differences in GI symptoms between all interventions. Fructans: overall GI symptoms and bloating significantly higher than gluten.
Picarelli et al. (2016) Italy [90]	RDBPC trial, NCGS patients **(*n* = 26)**	**GFD vs. GCD**, 1 day. (croissant with 10 g of gluten vs. gluten-free croissant)	No significant difference in overall symptom severity between gluten and placebo challenge.
Aziz et al. (2016) UK [91]	**Open label**, IBS-D patients **(*n* = 41)**	**GFD**, 6 weeks (information and advice GFD by dietician)	Decrease of symptoms in >70% of patients, significant after 2 weeks, similar results in HLA-DQ positive and negative
Elli et al. (2016) Italy [92]	RDBPC, **cross-over trial**, IBS patients with NCGS **(*n* = 98)**	**GFD vs. GCD**, 1 week per intervention, 1 week washout (gastro-soluble capsules with 5.6 g/day gluten powder or placebo). Run-in period of 3 weeks GFD.	14% of patients that responded to gluten withdrawal had symptomatic relapse during gluten challenge.
Shahbazkhani et al. (2015) Iran [93]	DB **RCT**, IBS patients **(*n* = 148)**	**GFD vs. GCD**, 6 weeks (packages with 52 g/day gluten powder, or rice starch as placebo). Run-in period of 6 weeks.	Significant improvement in overall symptom severity GFD (83.8%) vs. GCD (25.7%).
Di Sabatino et al. (2015) Italy [94]	RDBPC, **cross-over trial**, suspected NCGS patients **(*n* = 61)**	**GFD vs. GCD**, 1 week per intervention, 1 week washout. (gastro-soluble capsules with 4.4 g/day gluten vs. rice starch)	Significant increase in overall symptom severity during gluten compared to placebo. Abdominal bloating, pain and (extra)-intestinal symptoms significantly more severe during gluten-period.
Peters et al. (2014) Australia [95]	RDBPC, **cross-over trial** NCGS patients **(*n* = 22)**	**Gluten** (16 g/day) **vs. Whey** (16 g/day) **vs. Placebo**, 3 days per intervention, at least 3 days washout. (provided meals with 16 g/day whey protein vs. placebo)	No significant differences in GI symptoms between interventions. Significant more feelings of depression due to short-term exposure to gluten.
Vazquez-Roque et al. (2013) USA [96]	**RCT**, IBS-D patients **(*n* = 45)**	**GFD vs. GCD**, 4 weeks (standardized meals provided by metabolic kitchen, with or without gluten)	Significant increase in stool frequency GCD vs. GFD. Greater difference in HLA-DQ positive patients.
Biesiekierski et al. (2013) Australia [86]	(1) RDBPC, **cross-over trial**, IBS patients with NCGS **(*n* = 40)** **(2) Rechallenge**, IBS patients with NCGS **(*n* = 22)**	**(1) High gluten** (16 g/day) vs. **Low gluten** (2 g/day) vs. **Whey** (16 g/day), 1 week per intervention. **(2) Gluten** (16 g/day) vs. **Whey** (16 g/day) vs. Placebo (no additional protein), 3 days. Run-in period of 2 weeks, GFD and low FODMAP diet	Symptom improvement in all patients during run-in period (low FODMAP, gluten-free). Symptom deterioration in all groups, no specific gluten dose response.
Carroccio et al. (2012) Italy [97]	RDBPC, **cross-over trial**, suspected NCGS patients **(*n* = 920)**	**Wheat** (20 g/day) vs. **Xylose** (placebo), 2 weeks per intervention, at least 1 week washout. (gastro-soluble capsules). Elimination diet of 4 weeks prior to challenge.	Symptom improvement of at least 30% in wheat-free period (Salerno experts’ criteria): NCGS diagnosis was confirmed in 30% (*n* = 276) of subjects.
Biesiekierski et al. (2011) Australia [98]	**RCT**, IBS patients **(*n* = 39)**	**GFD vs. GCD**, 6 weeks (Muffin and bread with or without gluten, 16 g/day)	GCD baseline vs. 1 week: significant increase in overall symptom severity, as well as bloating, abdominal pain, tiredness, dissatisfaction with stool. GCD vs. GFD, 6 weeks: significant increase in severity of abdominal pain, tiredness and dissatisfaction with stool.

Abbreviations: DB: double-blind; FODMAPs: fermentable, oligo-, di-, monosaccharides and polyols; GCD: gluten-containing diet; GFD: gluten-free diet; GI: gastrointestinal; HLA-DQ: human leukocyte antigen-DQ; IBS: irritable bowel syndrome; IBS-D: irritable bowel syndrome with predominant diarrhea; NCGS: non-celiac gluten (or wheat) sensitivity; RCT: randomized, controlled trial; RDBPC: randomized, double-blind, placebo-controlled.

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
