# Peer review of "The Dietary Management of Patients with Irritable Bowel Syndrome: A Narrative Review of the Existing and Emerging Evidence"

_nutrients, 2019, doi:10.3390/nu11092162_

Round 1
Reviewer 1 Report
This narrative review discussed the evidenced for the dietary management of IBS. This article is well written and easy to follow. My only recommendation is to include a methods section. How were the included articles gathered? Were they retrieved via computerized databases, hand searches, or authoritative texts?
Author Response
This narrative review discussed the evidenced for the dietary management of IBS. This article is well written and easy to follow. My only recommendation is to include a methods section. How were the included articles gathered? Were they retrieved via computerized databases, hand searches, or authoritative texts?
Response: Thank you for this comment. We have added a methods section in the revised manuscript, where we described the search methodology.
Reviewer 2 Report
This is a comprehensive well written review with thorough critical evaluation of evidence (supported by well presented summary tables detailing the studies) and is a useful article for gastroenterologists and other health professionals that work with IBS.
I think more about methodology used for review should be included: did you use a systematic approach to literature searching for narrative review/synthesis, what were search terms, which databases were searched, inclusion exclusion criteria etc. ideally this shoudl be included in abstract and in method.
3 IgG
Refer to controversial aspects: some concern about laboratories being able to replicate results, related to frequency of consumption of food etc.
4 Fibre
Perhaps pertinent to mention different types of fibre here earlier in article and give brief definition? This would aid understanding for readers - mention of soluble/insoluble comes later on Line 185 but I think beneficial to mention it earlier and define differences.
Also consider fibre and short chain fatty acid production and link to microbiome and pH ?
5. Nice guidelines
Can you include comparison to any other national guidelines that consider dietary recommendations?
7. Lactose
Briefly mention Cows milk protein allergy - and its effects on gut/bowel - to enable differentiation.
9. Low carb diet
Give overview of what low carb diets are in % energy terms (can be as much as 40% energy from carbs) versus very low carb diets like the one mentioned of 20g a day.
10. Gluten
Line 395 - explain no known biomarkers for non coeliac gluten sensitivity.
11. Future Directions and recommendations
Line 453 - link to gut brain axis?
Author Response
This is a comprehensive well written review with thorough critical evaluation of evidence (supported by well presented summary tables detailing the studies) and is a useful article for gastroenterologists and other health professionals that work with IBS.
I think more about methodology used for review should be included: did you use a systematic approach to literature searching for narrative review/synthesis, what were search terms, which databases were searched, inclusion exclusion criteria etc. ideally this should be included in abstract and in method.
Response: Thank you for these comments. We have added a methods section in the revised manuscript, where we have described the search methodology.
IgG (3): Refer to controversial aspects: some concern about laboratories being able to replicate results, related to frequency of consumption of food etc.
Response: Thank you for this comment. We have added a brief sentence about this in paragraph 4.
Fibre (4): Perhaps pertinent to mention different types of fibre here earlier in article and give brief definition? This would aid understanding for readers - mention of soluble/insoluble comes later on Line 185 but I think beneficial to mention it earlier and define differences.
Also consider fibre and short chain fatty acid production and link to microbiome and pH ?
Response: Thank you for this comment. We have mentioned the definition of insoluble and soluble fibers in the first paragraph of this section. Next, we have clarified briefly the relation between fibers and short-chain fatty acids production by the gut microbiome.
Nice guidelines (5): Can you include comparison to any other national guidelines that consider dietary recommendations?
Response: Thank you for this comment. We have searched for other national guidelines with dietary recommendations for IBS patients but we did not find any suitable evidence based guidelines to compare with; hence, this has not been changed in the text.
Lactose (7): Briefly mention Cows milk protein allergy - and its effects on gut/bowel - to enable differentiation.
Response: This has now been added. We have added a short paragraph (3) where we have mentioned the gut physiology of cow’s milk protein allergy, and the difference between cow’s milk protein allergy and lactose intolerance.
Low carb diet (9): Give overview of what low carb diets are in % energy terms (can be as much as 40% energy from carbs) versus very low carb diets like the one mentioned of 20g a day.
Response: Thank you for your comment. We have revised this and the section about low-carbohydrates gives a better overview of what low and very-low carbohydrates mean in terms of energy amounts now.
Gluten (10): Line 395 - explain no known biomarkers for non coeliac gluten sensitivity.
Response: Thank you for this comment. We have added the clarification that there are no known biomarkers for non-celiac gluten sensitivity. The diagnosis can be confirmed by meeting the Salerno’s Expert criteria.
Future Directions and recommendations (11): Line 453 - link to gut brain axis?
Response: Thank you for this comment. We have added a sentence where we link the placebo and nocebo effect to the gut-brain axis.
Reviewer 3 Report
the revision work is excellent, widely discussed in each part and the cited references papers are all of recent works on the subject.
Author Response
The revision work is excellent, widely discussed in each part and the cited references papers are all of recent works on the subject.
Response: Thank you for this comment.